# Deep learning enables structured illumination microscopy with low light levels and enhanced speed

Luhong Jin[1,2,5], Bei Liu [1,5✉], Fenqiang Zhao[1,2], Stephen Hahn[1], Bowei Dong[1], Ruiyan Song[1], Timothy C. Elston[1,3], Yingke Xu [2,4✉] & Klaus M. Hahn [1✉]

Structured illumination microscopy (SIM) surpasses the optical diffraction limit and offers a two-fold enhancement in resolution over diffraction limited microscopy. However, it requires both intense illumination and multiple acquisitions to produce a single high-resolution image. Using deep learning to augment SIM, we obtain a five-fold reduction in the number of raw images required for super-resolution SIM, and generate images under extreme low light conditions (at least 100× fewer photons). We validate the performance of deep neural networks on different cellular structures and achieve multi-color, live-cell super-resolution imaging with greatly reduced photobleaching.

[1] Department of Pharmacology, University of North Carolina at Chapel Hill, Chapel Hill, NC 27599, USA. [2] Department of Biomedical Engineering, MOE Key Laboratory of Biomedical Engineering, Zhejiang Provincial Key Laboratory of Cardio-Cerebral Vascular Detection Technology and Medicinal Effectiveness Appraisal, Zhejiang University, 310027 Hangzhou, China. [3] Computational Medicine Program, University of North Carolina at Chapel Hill, Chapel Hill, NC 27599, USA. [4] Department of Endocrinology, The Affiliated Sir Run Run Shaw Hospital, Zhejiang University School of Medicine, 310016 Hangzhou, China. [5] These authors contributed equally: Luhong Jin, Bei Liu. ✉email: beiliu@email.unc.edu; yingkexu@zju.edu.cn; khahn@med.unc.edu

Structured illumination microscopy (SIM) is a widely used super-resolution technique that has had substantial impact because of its ability to double resolution beyond the light diffraction limit while using wide field illumination and maintaining compatibility with a wide range of fluorophores. SIM applies varying, nonuniform illumination on samples and then uses dedicated computational algorithms to derive super-resolution information from nine or fifteen sequentially acquired images, for 2D or 3D imaging, respectively. Since it was first introduced by the laboratories of Heintzmann[1] and Gustafsson[2] two decades ago, SIM has been evolving constantly to improve speed, resolution, and to decrease the required light dosages. Reconstruction algorithms have been developed to estimate microscope parameters robustly[3], minimize reconstruction artifacts[4,5], reduce the required number of raw images[6], and check the quality of the raw data and reconstruction[7]. The primary limitation of SIM is the need to obtain a series of high-quality images for each reconstructed high-resolution SIM image; this decreases temporal resolution and increases photobleaching.

Recently, there has been an explosion of deep-learning applications in many aspects of biological research. For microscopy, deep learning has demonstrated impressive capabilities in cell segmentation/tracking, morphology analysis, denoising, single molecule detection/tracking, and super-resolution imaging[8–10]. The use of deep learning for content-aware image restoration has shown great promise in denoising, enhancing signal-to-noise ratio, and isotropic imaging[11]. Deep neural networks have also been trained to increase the apparent magnification and resolution of images[12]. However, the potential of deep learning to increase the speed of SIM or to boost SIM's performance under low-light conditions has not been explored. We apply deep learning to increase the speed of SIM by reducing the number of raw images, and to retrieve super-resolution information from low-light samples. We accomplish this by reconstructing images using deep neural networks that have been trained on real images, enabling us to visualize specific complex cellular structures (mitochondria, actin networks etc.) and address complicated cellular or instrument-dependent backgrounds (e.g. out of focus light). The approach was named DL-SIM (deep learning assisted SIM).

## Results

**SIM reconstruction with fewer images.** U-Net is one of the most popular convolutional neural network architectures[13,14]. We show that U-Net can be trained to directly reconstruct super-resolution images from SIM raw data using fewer raw images. While conventional SIM typically requires nine or fifteen images for reconstruction, U-Net achieved comparable resolution with only three images. Also, using stacked U-Nets, we could restore high quality, high-resolution images from image sequences acquired using greatly reduced light dosages. We demonstrated the capabilities of DL-SIM in multi-color super-resolution imaging of living cells.

We first tested the ability of U-Net to reconstruct super-resolution images from SIM raw sequences. Conventional SIM excites specimens with sinusoidal waves at different angles and phases. Typically, it requires 9-images (3 angles, 3 phases) for two-beam SIM and 15 images (3 angles, 5 phases) for three-beam SIM[15]. We trained a single U-Net (U-Net-SIM15) by taking 15 SIM raw images as the input and the corresponding conventional SIM reconstruction results as the ground truth (Fig. 1a, Supplementary Fig. 1, Methods). U-Net-SIM15 was trained on four different subcellular structures: microtubules, adhesions, mitochondria and F-actin. Each dataset was randomly separated into subsets for training, validation, and testing respectively

(Supplementary Table 1). We tested performance on cells that had not been seen by the networks during the training step. U-Net-SIM15 consistently produced images with fidelity comparable to that of the conventional SIM algorithm (Fig. 1b, column U-Net-SIM15). We next tested whether we could accelerate SIM imaging by reducing the number of raw images for SIM reconstruction. We trained another U-Net (U-Net-SIM3) using only three SIM raw images (the first phase at each angle) as input, and again used the SIM reconstruction results from fifteen images as the ground truth. Surprisingly, U-Net-SIM3 could produce restored images (Fig. 1b, column U-Net-SIM3) with the quality of those produced using U-Net-SIM15 and the ground truth. We estimated the restoration error maps using SQUIRREL[16] for both conventional SIM reconstruction and U-Net output images against the average projection of the SIM raw data, and used both resolution-scaled error (RSE) and resolution-scaled Pearson coefficient (RSP) to quantify the quality of the restoration (Supplementary Fig. 2, Methods). Line profiles along microtubules, adhesions, mitochondria, and F-actin showed that U-Net reached a lateral resolution comparable to conventional SIM reconstruction (Fig. 1b). We also quantified the resolution of each approach using a recently reported approach based on decorrelation analysis[17] (Fig. 1c), peak signal-to-noise ratio (PSNR), the normalized root-mean-square error (NRMSE), and the structural similarity index (SSIM) (Supplementary Table 2, Methods). Furthermore, we applied the pre-trained model to visualize the dynamics of microtubules in living cells with high resolution (Supplementary Movie 1).

**SIM reconstruction from noisy input.** Next, we sought to increase acquisition speed and reduce photobleaching by using low laser power and reducing the exposure time (Supplementary Table 3). Reducing light levels degrades conventional SIM reconstruction because there is less information in the raw data. U-Net and other machine learning methods have been successfully adopted to recover information from low-light samples[11]. We therefore trained another U-Net (U-Net-SNR) to recover signals from poor-quality images, and fed the output from this to the pre-trained U-Net-SIM15. U-Net-SNR alone could recover information from low-light samples (e.g., periodic illumination patterns, Supplementary Fig. 3a, b). Combining the two networks produced good resolution throughout much of the images, but failed in some specific areas (Supplementary Fig. 3c). We hypothesized that training a deeper network by connecting two U-Nets could improve performance[11,18], so we constructed a new architecture (scU-Net) by chaining two U-Nets through skip-layer connections (Fig. 2a, Supplementary Fig. 4). The networks (both U-Net-SIM15 and scU-Net) were trained using an input of 15 SIM raw images obtained under low-light conditions, and using SIM reconstruction from normal light dosages as the ground truth. We found that both U-Net-SIM15 and scU-Net produced better restoration quality than conventional SIM reconstruction of low-light samples (Fig. 2b). We quantified network performance by calculating PSNR, NRMSE, and SSIM for the different reconstruction approaches relative to the ground truth (Supplementary Table 4). Our results showed that scU-Net provides the least restoration error (Supplementary Fig. 5, Row RSP and RSE in Supplementary Table 4). Both U-Net-SIM15 and scU-Net achieved higher resolution than conventional SIM reconstruction under low-light conditions, but scU-Net performed better on three out of four tested structures (Supplementary Table 5). We further tested the pre-trained scU-Net to visualize the dynamics of microtubules in living cells under extreme low-light conditions (Fig. 3a, Methods). The quality of the input and conventional SIM reconstruction was poor under

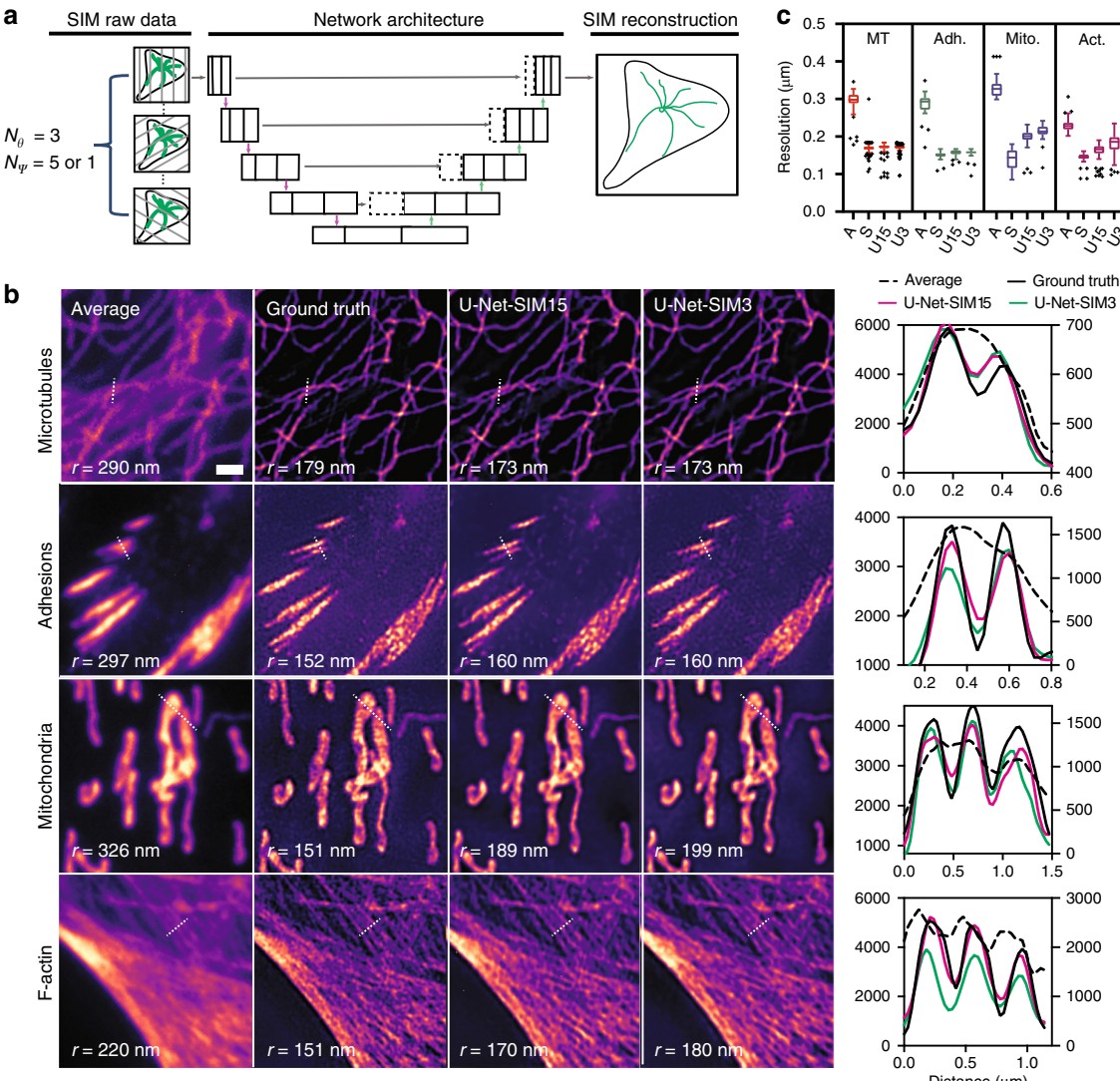

**Fig. 1 Super-resolution imaging with U-Net. a** Fifteen or three SIM raw data images were used as input and the corresponding SIM reconstructions from 15 images were used as the ground truth to train the U-Net. Θ: the angle of the sinusoidal patterned illumination; ψ: the phase of the patterned illumination. **b** Reconstruction results for different subcellular structures. Shown are average projections of 15 SIM raw data images (first column), the reconstruction results from a conventional SIM reconstruction algorithm (second column), U-Net-SIM15 output (third column), U-Net-SIM3 output (fourth column) and line profiles along the dashed line in each image (fifth column). In the line profile plot, the average is shown on the right $y$-axis and all others share the left $y$-axis. $r$ indicates the resolution. Shown are representative images randomly selected form the testing dataset indicated in Supplementary Table 1. The training datasets were collected from at least three independent experiments. **c** The achieved resolution of different approaches was estimated (Source data are provided as a Source Data file). MT microtubules ($n = 204$); Adh. adhesions ($n = 32$); Mito. mitochondria ($n = 61$); Act. F-actin ($n = 85$). A average; S SIM reconstruction; U15 U-Net-SIM15; U3 U-Net-SIM3. Tukey box-and-whisker plot shown with outliers displayed as dots (Methods). Scale bar: 1 μm.

these conditions (Fig. 3a, column 1 and 2). U-Net-SIM15 improved the reconstruction but missed some details (Fig. 3a, column 3). The scU-Net retrieved the missing structures from U-Net-SIM15 (Fig. 3a, column 4, white arrows), enabling us to track the dynamics of single microtubules (Fig. 3b, Supplementary Movie 2) with substantially reduced photobleaching (Supplementary Fig. 6). We used the pre-trained scU-Net to examine microtubule/mitochondrial interactions with short exposure time and low laser intensity (Fig. 3c, Supplementary Movie 3, Methods), but with no discernable compromise to image quality.

Using networks trained on synthetic tubular and point-like data, previous studies showed that U-Net could resolve sub-diffraction structures with at least 20× faster speed than super-resolution radial fluctuations (SRRF)[11,19]. We investigated the performance of U-Net trained on real biological samples. With a single U-Net (U-Net-SRRF5), we achieved comparable quality by taking as few as five total internal reflection fluorescence (TIRF) microscopy images, 40× fewer images than SRRF (Supplementary Fig. 7a–c). In general, 50–70 single cells were needed for the training step to obtain a working model (Supplementary Table 1). A model trained from a given intracellular structure produced significant artifacts when used to examine other structures (Supplementary Fig. 8). We applied transfer learning[20] to minimize the effort when adapting our pre-trained networks to other structures (Methods). U-Net-SIM15 pre-trained on microtubules was used to initialize a new U-Net retrained with other structures (adhesions, mitochondria and F-actin) (Methods). This achieved considerable improvement in the output quality even with only 200 training samples for each structure and 1/10 the training effort (Supplementary Fig. 9). Although we could achieve

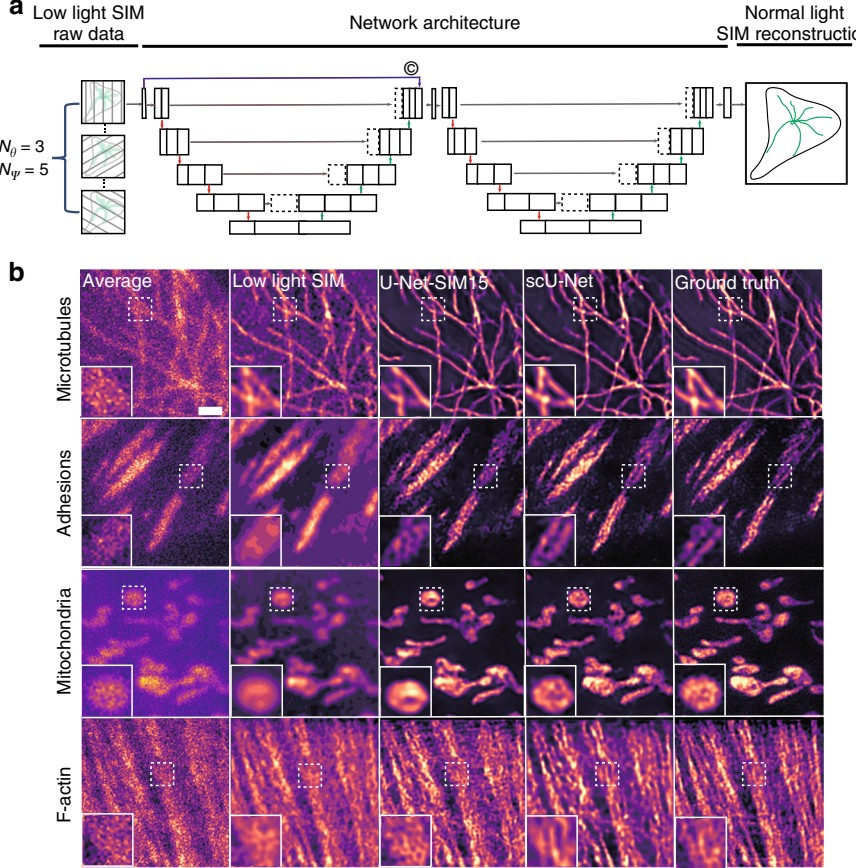

**Fig. 2 Super-resolution imaging under extreme low-light conditions. a** Two U-Nets were stacked through skip-layer connections. Fifteen SIM raw data images taken under low-light conditions were used as the input and the corresponding SIM reconstructions under normal-light conditions were used as the ground truth to train the scU-Net. **b** Reconstruction results for different subcellular structures (first row: microtubules; second row: adhesions; third row: mitochondria; fourth row: F-actin). Shown are average projections of 15 SIM raw data (first column), the reconstruction results from a conventional SIM reconstruction algorithm (second column), U-Net-SIM15 output (third column), scU-Net output (fourth column), and the ground truth from SIM reconstruction under normal-light conditions (fifth column). Shown are representative images randomly selected form the testing dataset indicated in Supplementary Table 1. The training datasets were collected from at least three independent experiments. The local enlargements show the restoration quality. Scale bar: 1 μm.

ultra-short exposure times (< 5 ms) for single frames, the minimum time interval between each processed image (1 s) was limited by our commercial SIM system, due to the time required to change the hardware between each image acquisition. Our data show that deep learning can substantially push this speed limit using home-built SIM systems[5,21] or faster commercial systems.

## Discussion

Here we use deep learning to produce high-quality SIM images with fewer input images and with lower intensity and/or shorter exposure. Importantly, we quantitatively showed that deep learning can achieve resolution comparable to that of conventional SIM reconstruction algorithms (Fig. 1c). We demonstrated that by taking SIM raw data as input, we could preserve patterned illumination information when recovering signals from low-light samples (Supplementary Fig. 3). This could be useful for other advanced imaging techniques that utilize structured illumination, such as repetitive optical selective exposure (ROSE)[22] and SIM-FLUX[23]. These techniques use structured illumination to boost precision of single molecule imaging, requiring the precise estimation of the illumination phase. This would be impossible using a single TIRF image or average SIM raw data as input. Finally, previous studies by the Ozcan lab[12] used Generative Adversarial Nets (GANs), to convert TIRF images into SIM-quality images.

We are using U-Net, a convolutional neural network. GANs is a competitive process between the generator (G) and discriminator (D). Therefore, two networks have to be trained and the loss between G and D has to be balanced carefully. GANs perform well in image-to-image translation, but are generally difficult to train[24–26], requiring more input images and more training epochs[12]. Our method only required 50–70 samples per structure and training for 2000 epochs. The U-Nets based approach is therefore more user friendly, especially for biologists and users who are inexperienced with deep learning.

## Methods

**Microscopes**. SIM imaging was performed on the Nikon N-SIM system equipped with 488 nm (70 mW), 561 nm (70 mW), and 647 nm (125 mW) laser lines, two EMCCD cameras (Andor iXon3) and a 100×, NA 1.45 objective (Nikon, CFI Apochromat TIRF 100XC Oil). For the training datasets, we used 10% intensity of 488/561/647 nm lasers and 200 ms exposure time to acquire SIM images with high quality, and 1% intensity of 488/561/647 nm lasers and 20 ms exposure time to acquire SIM images under low-light conditions. For the live-cell microtubule imaging experiments, we used 10% intensity of the 561 nm laser with 100 ms exposure for normal light conditions, and used 1% intensity of the 561 nm laser and 5 ms exposure for low-light conditions. For the dual-color experiments, we used the 488 nm laser at 2% power and the 561 nm laser at 1% power with 50 ms exposure time to visualize microtubule–mitochondria interactions.

The SRRF imaging was performed on a home-built TIRF microscope equipped with 488, 561, and 647 nm laser lines, two sCMOS cameras (Photometrics, Prime 95B) and a 100×, NA 1.49 TIRF objective (Olympus, UAPON100XOTIRF). For

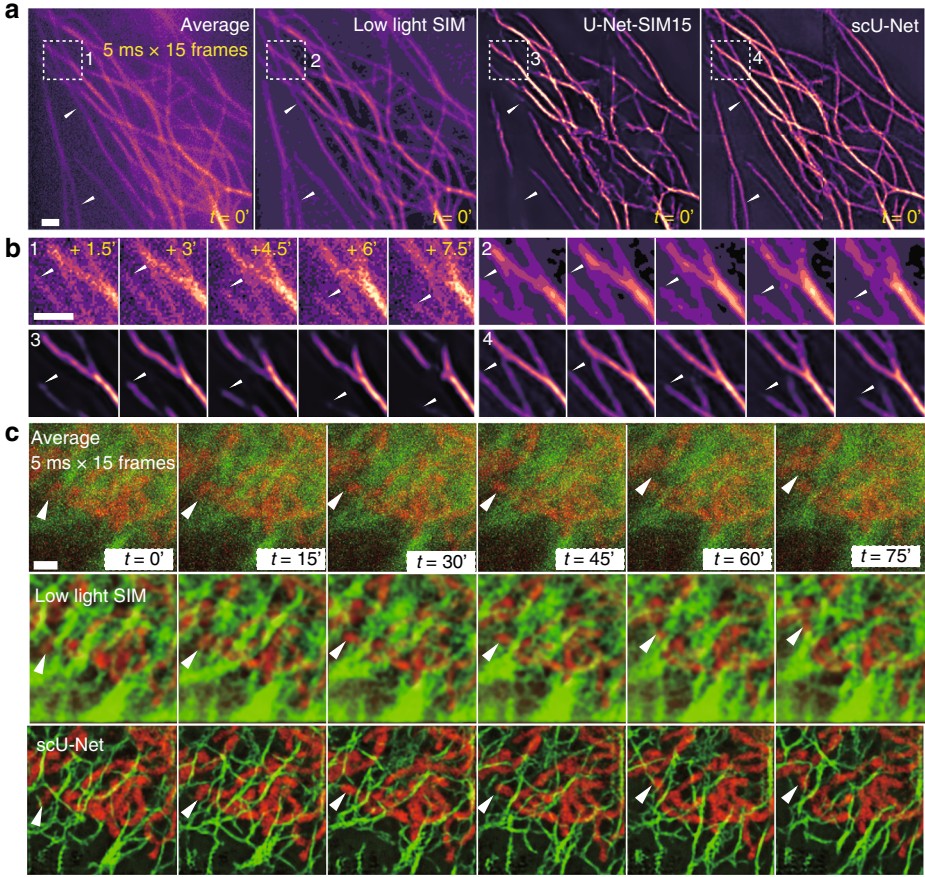

**Fig. 3 scU-Net for live-cell imaging. a** Reconstruction results for microtubules in living cells. A representative time point is shown. The missing structures from U-Net-SIM15 were recovered by scU-Net (white arrows). First panel shows the average projections of 15 SIM raw images; second panel shows the SIM reconstruction; third panel shows the U-Net-SIM15 output; fourth panel shows the scU-Net output. $n = 3$, from three independent experiments. **b** Enlarged views of areas indicated by the white-dashed box in **a** are shown. The dynamics of a single microtubule (white triangle) was well restored by scU-Net. **c** scU-Net reveals the dynamics of microtubule–mitochondria interactions. First row: average projections of 15 SIM raw images; second row: SIM reconstruction; third row: scU-Net output. $n = 3$, from three independent experiments. Scale bar: 1 μm.

imaging microtubules, we used 0.6 mW from a 647 nm laser and 100 ms exposure time.

**Cell culture and preparation**. For the training step, fixed cell samples were used. We prepared four training datasets from various subcellular structures, including microtubules, adhesions, mitochondria, and F-actin. For imaging mitochondria and F-actin, fluorescently prelabeled commercial slides (Molecular Probes, F36924) were used. Microtubule and adhesion samples were prepared as follows:

Microtubules: Mouse embryonic fibroblast (MEF) cells were fixed with −20 °C 100% methanol for 3 min. The cells were washed for five times with 0.1% Triton-X100 in phosphate-buffered saline (PBS), and then permeabilized with 0.5% Triton-X100 in PBS for 10 min. Next, the cells were washed three times again and blocked for 15 min with blocking buffer (5% BSA in the wash buffer). The cells were incubated with beta-tubulin antibody (Developmental Studies Hybridoma Bank, E7) (1:500 dilution into blocking buffer), followed by the Anti-rabbit IgG (H + L) F(ab')2 Fragment conjugated with Alexa-647 dye (CST, 4414S).

Adhesions: MEF cells were fixed with 37 °C 4% paraformaldehyde for 10 min at room temperature. The cells were washed with PBS three times and blocked with 3% BSA, 0.2%Triton-X100 in PBS for 30 min. The cells were incubated with 1:100 diluted primary antibody (Santa Cruz Biotechnology, sc-365379) for 30 min and washed (0.2% BSA, 0.1% Triton-X100) five times, followed by the staining of the anti-rabbit IgG (H + L), F(ab')2 fragment conjugated with Alexa-647 dye (CST, 4414S).

Live cells labeling: COS-7 cells were cultured in Dulbecco's modified Eagle's medium (Fisher Scientific, 15-013-CV) supplemented with 10% fetal bovine serum (Gemini, 100-106) and 5% GlutaMax (Thermo Fisher, 35050061). To visualize microtubules, COS-7 cells were transfected with a microtubule-associated protein (EMTB-3xmCherry or EMTB-3xEGFP), using Viromer Red transfection reagent (OriGene Technologies, TT1003102). Mitochondria were stained with MitoTracker Green FM (Thermo Fisher, M7514) according to the manufacturer's instructions.

**Data preprocessing**. For deep learning, the size of the training dataset should be as large as possible to cover the distribution of images in the task domain. However, collecting huge amount of single cell data can be time consuming and expensive. We therefore cropped the original image stacks into smaller patches to generate more training samples for all the experiments. In SIM experiments, the size of the raw image stack was $512 × 512 × 15$ (width × height × frame). To prepare the input for U-Net-SIM15, the raw stack was cropped into $128 × 128 × 15$ patches. For U-Net-SIM3, only the first phase of three illumination angles were selected, producing $128 × 128 × 3$ patches. After that, we manually discarded the patches, which contained only background information. We then located the same areas on the SIM reconstruction images to produce the corresponding ground-truth images. In total, we obtained 800–1500 samples for different structures, which were then randomly divided into training, validation, and testing subsets. Detailed information about each dataset is in Supplementary Table 1.

Normalization is also important to the efficiency and robustness of the network. We normalized the input images to the maximum intensity of the whole input dataset and the ground truth images to the maximum intensity of the SIM reconstruction dataset.

Since U-Net requires the width and height of the input images to match the ground truth images, we resized the input dataset to $256 × 256 × C$ ($C$ means the number of channels for the input data; it differs among different experiments) using bicubic interpolation.

**Network architectures and training details**. U-Net-SIM15 and U-Net-SIM3, U-Net-SNR, and U-Net-SRRF share similar network architectures (Supplementary Fig. 1) and they only differ in the numbers of channels of either input or output (ground truth) dataset (U-Net-SIM15: $C_{in} = 15$, $C_{out} = 1$; U-Net-SIM3: $C_{in} = 3$, $C_{out} = 1$; U-Net-SNR: $C_{in} = 15$, $C_{out} = 15$; U-Net-SRRF: $C_{in} = 5$, $C_{out} = 1$; $C_{in}$ and $C_{out}$ are the numbers of channels of the input and output, respectively). For the U-Net-SNR, we took 15 SIM raw data images acquired under low-light conditions as the input and the same sample under normal light conditions as the ground truth.

For the U-Net-SRRF, we used five TIRF frames as the input and the SRRF reconstruction from 200 frames as the ground truth. For the SIM experiment under low-light conditions, the scU-Net was used (Supplementary Fig. 4). The training details for each experiment are listed in Supplementary Table 1. The loss function for all experiments is defined as:

$$\text{loss} = \frac{1}{W \times H} \times \left( \sum_{i=1}^{W} \sum_{j=1}^{H} (U(i,j) - V(i,j)) + 5 \times \sum_{i=1}^{W} \sum_{j=1}^{H} (U(i,j) - V(i,j))^2 \right). \tag{1}$$

Here $W$ and $H$ represent the width and height of the ground truth image in the training step ($W = 256$, $H = 256$ across all networks related to SIM and $W = 320$, $H = 320$ for the SRRF experiment). $U$ and $V$ represent the ground truth image and the output from the network, respectively. The codes for training and testing were written using Python with PyTorch framework. All source codes will be available online (https://github.com/drbeiliu/DeepLearning).

**Quantification of performance for each network.** For the testing part, we used four metrics to evaluate the performance of our networks, including image resolution, PSNR, NRMSE, and SSIM. The resolution of each cropped image was estimated using the ImageDecorrleationAnalysis plugin in Fiji/ImageJ with the default parameter settings[17]. Note that for low-light images, the image quality was so poor that the plugin failed to report a reasonable value. In that case, we used the whole-cell image, instead of the cropped patches to estimate the resolution. As for PSNR, NRMSE, and SSIM, we used the SIM reconstruction results under normal-light conditions as the ground truth. Each metric was calculated as below:

$$\text{PSNR} = 20 \times \log_{10} \left( \frac{255}{\sqrt{\sum_{i=1}^{W} \sum_{j=1}^{H} (U(i,j) - V(i,j))^2 / (W \times H)}} \right). \tag{2}$$

$$\text{NRMSE} = \frac{\sqrt{\sum_{i=1}^{W} \sum_{j=1}^{H} (U(i,j) - V(i,j))^2 / (W \times H)}}{\sigma_U}. \tag{3}$$

$$\text{SSIM} = \left( \frac{2\bar{U}\bar{V} + C_1}{\bar{U}^2 + \bar{V}^2 + C_1} \right) \times \left( \frac{2\sigma_{UV} + C_2}{\sigma_U^2 + \sigma_V^2 + C_2} \right). \tag{4}$$

Here $W$ and $H$ represent the width and height of the ground truth image in the training step ($W = 256$, $H = 256$ across all networks). $U$ and $V$ represent the ground truth image and the output of the network, respectively. $\bar{U}$ and $\bar{V}$ represent the averages of $U$ and $V$. $\sigma_U$ and $\sigma_V$ represent the variances of $U$ and $V$. $\sigma_{UV}$ is the covariance of $U$ and $V$. The items $C_1$ and $C_2$ are small positive constants that stabilize each term ($C_1 = (k_1 L)^2$, $C_2 = (k_2 L)^2$, $L$ is the dynamic range of the pixel-values, $k_1 = 0.01$ and $k_2 = 0.03$ by default). The code for calculating the performance was written with Python.

We then computed the performance of each metric for each architecture based on the output of the networks and the ground truth images (Supplementary Table 2, Supplementary Table 4). RSP and RSE were introduced before to assess the quality of super-resolution data[16] and were calculated using NanoJ-SQUIRREL (https://bitbucket.org/rhenriqueslab/nanoj-squirrel/wiki/Home).

**Transfer learning.** Directly applying a model trained on one specific structure to other structures may produce significant artifacts (Supplementary Fig. 8), which means that each target needs a unique model. In theory, we need to prepare ~1000 training samples and train the network for 2–3 days (~2000 epochs) on a consumer-level graphics card (NVIDIA GTX-1080 GPU) to get a working model for each structure we tested. We adopted transfer learning[20] to reduce the effort of imaging new structures. Briefly, we took the parameters obtained from a pre-trained network to initialize a new network and started retraining on a different structure with smaller training samples size (200 of cropped patches). We validated the effectiveness of transfer learning in restoring different structures. Even with reduced training efforts (200 epochs), the new model produced results comparable to the model trained with a much larger dataset and greater training effort (Supplementary Fig. 9).

**SRRF experiment.** In the SRRF experiment, the original input images were cropped into $64 \times 64 \times 5$ (width × height × frame), and the original ground truth images, which were computed from 200 TIRF images, were cropped into $320 \times 320 \times 1$. Note that the first 5 TIRF images were used from the total of 200 TIRF images. Since the size of the SRRF super-resolution image is larger than the input, we resized the cropped input image ($64 \times 64 \times 5$) into $320 \times 320 \times 5$ using bicubic interpolation to match the size of the ground truth.

**Statistical analysis.** In Fig. 1c, we used a Tukey box-and-whisker plot generated by GraphPad Prism 8.0. The box extends from the 25th and 75th percentiles and the line in the middle of the box indicates the median. To define whiskers and outliers, the inter-quartile distance (IQR) is firstly calculated as the difference between the 25th and 75th percentiles. The upper whisker represents the larger

value between the largest data point and the 75th percentile plus 1.5 times the IQR; the lower whisker represents the smaller value between the smallest data point and the 25th percentile minus 1.5 times the IQR. Outliers (any value larger than the upper whisker or smaller than the lower whisker) are displayed as dots. P values provided in Source Data are obtained from a Student's paired t test, with a two-tailed distribution.

**Reporting summary.** Further information on research design is available in the Nature Research Reporting Summary linked to this article.

## Data availability
All relevant data are available within the article and the Source Data. The training datasets are available from the corresponding author upon request, due to size limitations. The code and the pre-trained networks are available on GitHub: https://github.com/drbeiliu/DeepLearning and on http://hahnlab.com/tools/. The source data underlying Fig. 1c, Supplementary Tables 2, 4 and 5 are provided as a Source Data File.

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

## Acknowledgements

This work was funded by the NIH (R35GM122596 to K.M.H., R35GM127145 to T.C.E.), the National Key Research and Development Program of China (2018YFE0119000 and 2016YFF0101406 to Y.X.) and the Zhejiang Provincial Natural Science Foundation (LR18H180001 to Y.X.). We thank Tony Perdue from the Department of Biology Microscopy Core at UNC for assistance with SIM imaging.

## Author contributions

B.L., L.J., and K.M.H. conceived the project. L.J. and B.L. performed the imaging. L.J. and F.Z. developed the code with the help of B.D., R.S., and S.H. L.J. prepared the data and figures with the help of all authors. The manuscript was written with input from all authors, starting from a draft provided by B.L. T.E. provided oversight of mathematical aspects of the project, and Y.X. provided insight into imaging and microscopy. B.L., Y.X., and K.M.H. supervised the project.

## Competing interests

The authors declare no competing interests.
