## [Peer Review File · Nature Communications]

Reviewers' comments:

Reviewer #1 (Remarks to the Author):

In this manuscript the authors present a deep learning approach to reconstruct images of structured illumination microscopy, especially under low light conditions. The authors demonstrated U-net based networks and variants that are capable of reconstructing SIM images.

The authors seem to have missed previous demonstrations of deep learning based SIM reconstructions in the literature. Deep learning-based SIM image reconstruction and super-resolution have been previously reported in *Nat Methods* 16, 103–110 (2019).

Overall this reviewer finds the current submission more suitable for a specialized journal as it does not have the conceptual or technical advances expected from a high impact journal such as *Nature Communications*.

Some points that the authors can find useful for a future submission of their work are listed below.

1. The network output images do not always match well with the ground truth images. For example, in the third row in Fig. 2(b), although both network models generate more observable details, none of them are confirmed by a ground truth image. In the fourth column in Fig 2(b), the filamentous structures in U-Net-SIM15, scU-Net, and ground truth images are all different.
2. The low light performance was demonstrated in Fig. 3. However, without ground truth images compared side-by-side, it is difficult to understand the deep network's efficacy.
3. A thorough discussion of the authors results against *Nat Methods* 16, 103–110 (2019) would be useful for the readers, and is essential for proper practice of scholarship.

Reviewer #2 (Remarks to the Author):

Jin et al. present a machine learning approach to structured illumination microscopy reconstruction. Overall this is an interesting paper and would contribute significantly to the field, allowing reconstructions with lower S:N input data, or fewer input images helping with photobleaching effects which frequently limit super resolution imaging experiments, especially in live cells.

Overall I think the paper would be publishable in the following issues were addressed.

- 1) The authors fail to credit Heintzmann & Cremer who had very similar ideas to Gustaffson at around the same time in developing SIM fluorescence imaging.

Heintzmann, R.; Cremer, C. Laterally modulated excitation microscopy: improvement of resolution by using a diffraction grating *Proc. SPIE* 1998, 3568, 185– 196

- 2) They state that they reduce the laser power by 10x and the exposure time by 20x for the low light imaging, however they have several different conditions. One for fixed cells, one for the single colour time lapse and third for the dual colour time lapse, they need to properly describe what they are referring to, and make this agree with their abstract where they say 100x fewer photons.

- 3) The authors state "We found that scU-Net produced better

restoration quality than either SIM reconstruction or U-Net-15" (presumably the model referred to as U-Net-SIM15 elsewhere). I assume this refers to the low-light SIM reconstruction as the "normal" light level is used as the ground truth, this needs to be clarified.

4) The authors say that scU-Net achieved considerable improvements over other approaches, yet table 3 clearly shows it is no better than U-Net-SIM15. I think they overstate the improvement from this model.

5) A few sentences later they say "...scU-net produced the best results..." How is this defined? The data in the paper shows it is no better than the U-Net-15?

6) I don't see how the small section on SRRF processing adds to the paper at all. The paper is almost entirely about SIM processing and this section seems unnecessary.

7) I wonder how edge effects on their small training tiles (128x128) influence the error levels in the ML reconstructions. They don't say anything about this, yet the ground truth data is from 512x512 images and the models are trained using subsections to try and enable model training with fewer input image stacks.

8) In the SRRF processing section they say they trained the model with 5 input images out of 200, but don't say which 5. The first 5, or ones either randomly or regularly spaced within the 200 frames.

9) The authors say in the methods section they used 4 metrics to compare models but only report on 3, they don't give any data about the resolution measured by the imageJ plugin.

Reviewer #1 Comments:

“In this manuscript the authors present a deep learning approach to reconstruct images of structured illumination microscopy, especially under low light conditions. The authors demonstrated U-net based networks and variants that are capable of reconstructing SIM images.

The authors seem to have missed previous demonstrations of deep learning based SIM reconstructions in the literature. Deep learning-based SIM image reconstruction and super-resolution have been previously reported in Nat Methods 16, 103–110 (2019).

Overall this reviewer finds the current submission more suitable for a specialized journal as it does not have the conceptual or technical advances expected from a high impact journal such as Nature Communications.

Some points that the authors can find useful for a future submission of their work are listed below.”

Response:

We thank the reviewer for bringing this published work to our attention, and regret not citing it in the original draft. The paper from the Ozcan lab¹ shows a promising approach to convert low magnification images to high magnification images (e.g. 10 X to 20 X), and low-resolution images to high-resolution images (confocal to stimulated emission depletion). Converting a TIRF image to a SIM-compatible image is also considered briefly. Our work differs from Ozcan in four ways:

Firstly, the motivation for our paper and Ozcan’s paper are very different. Ozcan takes a single TIRF image (or the average of nine SIM raw images) as the input and converts it to a SIM-quality image. Their motivation is to improve the quality of TIRF microscopy. Our goals are to speed up SIM experiments and to reduce photo-toxicity while maintaining high resolution. By using deep learning, our method requires fewer images and lower light excitation. Importantly, we quantitatively showed that

deep learning could achieve resolution comparable to that of conventional SIM reconstruction algorithms (Fig. 1c).

Second, in conventional SIM imaging, the patterned illumination extends the spatial frequencies in the raw data^{2,3}. We hypothesize that simply taking single TIRF image or the average of SIM raw images as input would lose high frequency information, thus hurting the achievable quality of the network output (detailed comparison results are shown on Table 1 below). We therefore trained a new network by taking the average of the SIM raw images (pseudo TIRF) as input and the conventional SIM reconstruction as ground truth (named as U-Net-TIRF). Although we saw improvements over pseudo TIRF images, U-Net-TIRF did not surpass the performance of U-Net-SIM15 (see Table 1 underneath).

Table 1. Evaluation of the performance of U-Net-TIRF.

	Structures	Average	U-Net-TIRF	U-Net-SIM15
PSNR	microtubules	13.71±1.83	33.39±5.64	33.69±5.20*
	adhesions	16.29±1.85	24.85±1.99	27.81±1.59*
NRMSE	microtubules	1.76±0.62	0.20±0.11	0.19±0.10*
	adhesions	1.08±0.32	0.39±0.09	0.28±0.06*
SSIM	microtubules	0.27±0.10	0.96±0.05	0.96±0.04*
	adhesions	0.32±0.08	0.79±0.04	0.84±0.05 *

Note: asterisk indicates the best performance of each row. Average: the average of fifteen SIM raw images.

Third, we demonstrated that by taking SIM raw data as input, we could preserve patterned illumination information when recovering signals from low light samples (Supp. Figure 3). This could be useful for other advanced imaging techniques that utilize structured illumination, such as repetitive optical selective exposure (ROSE)⁴ and SIMFLUX⁵. These techniques use structured illumination to boost precision of single molecule imaging, requiring the precise estimation of the illumination phase. This would be impossible using a single TIRF image or average SIM raw data as input.

Forth, Ozcan's work uses Generative Adversarial Nets (GANs), while we are using U-Net, a convolutional neural network. GANs is a competitive process between the generator (G) and discriminator (D). Therefore, two networks have to be trained and the loss between G and D has to be balanced carefully. GANs perform well in image-to-image translation, but are generally difficult to train⁶⁻⁸. This is also reflected by the fact that in Ozcan's work, 3003 samples were used for TIRF-SIM application and 50,000 epochs were trained to obtain a working model. Our method only required 50-70 samples (~100X less) per structure and trained for 2,000 epochs (25X less). We believe that our U-Nets based approach is more user friendly, especially for biologists and users who are inexperienced with deep learning.

We have summarized these points in the revised manuscript (page 4, paragraph 2).

1. (Figure 2) *"The network output images do not always match well with the ground truth images. For example, in the third row in Fig. 2(b), although both network models generate more observable details, none of them are confirmed by a ground truth image. In the fourth column in Fig 2(b), the filamentous structures in U-Net-SIM15, scU-Net, and ground truth images are all different."*

Response:

It is true that the outputs from the networks were not as good as the ground truth for some subcellular structures. However, here our goal was to recover signals under low light conditions, surpassing the performance of conventional SIM. For this purpose, our models were validated by the examples shown in Figure 2, by the relative error maps in Supplementary Fig. 5, and by the parameters calculated in Supplementary Table 4.

It is worth noting that the output of any deep neural network is a prediction, which means it comes with uncertainty or error. The error exists in all previously published studies and has been well discussed⁹⁻¹³. As we mention in the paper,

reducing laser light dosages to 1% of normal light dosage degrades conventional SIM reconstruction because there is less information in individual raw images. Reducing light intensity will also affect the performance of the U-Net and any other potential algorithms. An appropriate way to evaluate the performance for computational algorithms is to compare the reconstruction results directly with the ground truth. Higher similarity indicates better reconstruction results. Therefore, the calculated values in Supplementary Table 4, we conclude that our network does have the potential to accurately extend SIM to low light imaging conditions. The error maps in Supplementary Fig. 5 also present the precision of different approaches while producing superresolution images from low quality images.

2. (Figure 3) *“The low light performance was demonstrated in Fig. 3. However, without ground truth images compared side-by-side, it is difficult to understand the deep network’s efficacy.”*

Response:

Figure 3 shows microtubule dynamics imaged using extreme low light intensity and short exposure times. This was used to improve the temporal resolution of the imaging, and to decrease photobleaching. It was not possible to obtain ground truth images with the same temporal resolution and for the same time period, precisely because conventional SIM could not produce high quality images with such short exposure times and low light. The underlying message we want to convey is that although the networks are trained with fixed cell samples, researchers can directly apply them to live cell studies. The performance of the neural networks is evaluated carefully in Figure 2, Supplementary Fig. 5 and Supplementary Table 4. By comparing the microtubules reconstructed by different methods, one can appreciate the capability of scU-Nets in improving image quality and signal-to-noise ratio. To better illustrate the striking differences, we have added a new image gallery in Figure 3c (top), showing the average of the input images for comparison.

3. (Page 2, paragraph 1) *“A thorough discussion of the authors results against Nat*

Methods 16, 103–110 (2019) would be useful for the readers, and is essential for proper practice of scholarship.”

Response:

We thank the reviewer for pointing out our oversight. We have cited the work from Ozcan’s group and have compared the two approaches in (page 4, paragraph 2).

Review #2 Comments:

“Jin et al. present a machine learning approach to structured illumination microscopy reconstruction. Overall this is an interesting paper and would contribute significantly to the field, allowing reconstructions with lower S:N input data, or fewer input images helping with photobleaching effects which frequently limit super resolution imaging experiments, especially in live cells.

Overall I think the paper would be publishable in the following issues were addressed.”

Response:

We thank the reviewer for this positive assessment of the manuscript and for all their constructive comments. Below we provide detailed responses to each comment:

1. (Page 1) *“The authors fail to credit Heintzmann & Cremer who had very similar ideas to Gustaffson at around the same time in developing SIM fluorescence imaging. Heintzmann, R.; Cremer, C. Laterally modulated excitation microscopy: improvement of resolution by using a diffraction grating Proc. SPIE 1998, 3568, 185–196”*

Response:

We apologize for this oversight. The pioneering work of Heintzmann & Cremer has now been included.

2. (Page 3, 2nd paragraph) *“They state that the reduce the laser power by 10x and the exposure time by 20x for the low light imaging, however they have several different conditions. One for fixed cells, one for the single colour time lapse and third for the dual colour time lapse, they need to properly describe what they are referring to, and make this agree with their abstract where they say 100x fewer photons.”*

Response:

The original text was confusing, and we have adjusted these statements in both the abstract and page 3, paragraph 2 of the revised manuscript. We have also added a new Supplementary Table 3 to summarize the imaging conditions for different experiments.

3. (Page 3, 2nd paragraph) *“The authors state “We found that scU-Net produced better restoration quality than either SIM reconstruction or U-Net-15” (presumably the model referred to as U-Net-SIM15 elsewhere). I assume this refers to the low-light SIM reconstruction as the “normal” light level is used as the ground truth, this needs to be clarified.”*

Response:

We apologize for the misunderstanding. Here SIM reconstruction means SIM reconstruction of low light samples and U-Net-15 refers to U-Net-SIM15. We have corrected this in our revised manuscript.

4. (Page 3, 2nd paragraph) *“The authors say that scU-Net achieved considerable improvements over other approaches, yet table 3 clearly shows it is no better than U-Net-SIM15. I think they overstate the improvement from this model.”*

Response:

Thank you for your helpful suggestion. We have modified this in the revised manuscript. From the overall evaluation (PSNR, NRMSE and SSIM), their

performances were similar and scU-Net achieved further improvement on three out of four tested structures (Supplementary Table 4). Under low light conditions, both networks (U-Net-SIM15 and scU-Net) achieve better resolution than traditional SIM reconstruction algorithms (Supplementary Table 5). With three out of four tested structures, scU-Net achieved comparable or even better resolution than U-Net-SIM15. In addition, from the error maps shown on Supplementary Figure 5 and the RSP/RSE measurements in Supplementary Table 4, we conclude that the deeper network (scU-Net) gives a better prediction than U-Net-SIM15. We noticed that in live cells, the sc-Uet method performs better than U-Net-SIM15, as is shown on both Figure 3a and Supplementary Video 2.

5. (Page 3) *"A few sentences later they say "...scU-net produced the best results..." How is this defined? The data in the paper shows it is no better than the U-Net-15?"*

Response:

This is a valuable correction. In most cases scU-Net performs better than U-Net-SIM15 but not always. We have therefore removed the phrase "... the scU-Nets produced the best results...". From Fig. 2b and Fig. 3a, U-Net-SIM15 and scU-Net achieved better resolution than low light SIM, while the results from scU-Net contained more detail and had less errors (see white arrows in Fig 3a). We are more precise in our conclusions in the revised manuscript.

6. (Page 4) *"I don't see how the small section on SRRF processing adds to the paper at all. The paper is almost entirely about SIM processing and this section seems unnecessary."*

Response:

SRRF is a popular, computational based super-resolution technique. Previously, CARE¹⁰ demonstrated the great promise of using deep learning to accelerate SRRF. The network was trained with synthetic data. We thought our readers would like to know how efficient it would be using real samples and how large a training dataset is

needed. Although it is not the focus of this paper, we would like to leave these results in the manuscript.

7. (Page 8, Online methods. Data preprocessing.) *“I wonder how edge effects on their small training tiles (128x128) influence the error levels in the DL reconstructions. They don't say anything about this, yet the ground truth data is from 512x512 images and the models are trained using subsections to try and enable model training with fewer input image stacks.”*

Response:

We thank the reviewer for bringing up this interesting question. We cropped the images for two reasons: first, to enlarge the size of the datasets; second, to be able to run the training on a single modern GPU (limited by the memory). In general, the patch size should be larger than the targets¹⁴. The organelles in this study (cytoskeletons, mitochondria and adhesions) are all much smaller than our chosen patch size (128 X 128). Smaller patch size (64 X 64) has also been used before (e.g. Ozcan's paper). From the error maps (Supp. Fig 2 and Supp. Fig 5), we did not observe significant errors along the edges. To better address the reviewer's concerns, we re-trained the U-Net-SIM15 network with larger patch size (200 X 200 and 256 X 256) on adhesions samples. We did not see a significant improvement over our old networks. On the contrary, the performance dropped, likely due to insufficient amount of training samples (see Table below). In practice, to process a larger image (e.g. 512 X 512), one could scan overlapped patches across the image and take the center of each patch for stitching to minimize the edge effect. This also has been implemented in our code on GitHub.

Table 2. Quantitative analysis of the performance of U-Net-SIM15 with different patch sizes. Networks were trained on adhesions.

Patch size	Average			U-Net-SIM15		
	128 X 128	200 X 200	256 X 256	128 X 128	200 X 200	256 X 256
PSNR	16.29±1.9*	21.38±3.4	21.36±3.2	27.81±1.6*	26.40±2.9	25.77±4.1

NRMSE	1.08±0.32*	0.79±0.30	0.83±0.27	0.28±0.06*	0.45±0.17	0.55±0.36
SSIM	0.32±0.08*	0.68±0.09	0.67±0.1	0.84±0.05*	0.77±0.08	0.77±0.11

Note: asterisk indicates the best performance of each row.

8. (Page 4) *“In the SRRF processing section they say they trained the model with 5 input images out of 200, but don't say which 5. The first 5, or ones either randomly or regularly spaced within the 200 frames.”*

Response:

For the SRRF processing, we used the first 5 TIRF images as the input for the network. This has now been noted in the Methods section of the revised manuscript.

9. (Page 7, Online methods. Quantification of performance for each network.) *“The authors say in the methods section they used 4 metrics to compare models but only report on 3, they don't give any data about the resolution measured by the imageJ plugin.”*

Response:

For U-Net-SIM15 and U-Net-SIM3, the resolution measurements were shown on the revised new Figure 1c. In the revised manuscript, we have added a new Supplementary Table 5, which summarized the resolution measurements by U-Net-SIM15 and scU-Net methods. We have also modified the Supplementary Table 2 and Supplementary Table 4, to include new data on RSP and RSE measurements.

Bibliography

1. Wang, H. *et al.* Deep learning enables cross-modality super-resolution in fluorescence microscopy. *Nat. Methods* **16**, 103–110 (2019).
2. Heintzmann, R. & Cremer, C. G. Laterally modulated excitation microscopy: improvement of resolution by using a diffraction grating. *BiOS Europe'98* (1999).
3. Gustafsson, M. G. Surpassing the lateral resolution limit by a factor of two using structured illumination microscopy. *J. Microsc.* **198**, 82–87 (2000).
4. Gu, L. *et al.* Molecular resolution imaging by repetitive optical selective exposure. *Nat. Methods* **16**, 1114–1118 (2019).
5. Cnossen, J. *et al.* Localization microscopy at doubled precision with patterned illumination. *Nat. Methods* (2019). doi:10.1038/s41592-019-0657-7
6. Salimans, T. *et al.* Improved Techniques for Training GANs. (2016).
7. Creswell, A. *et al.* Generative adversarial networks: an overview. *IEEE Signal Process. Mag.* **35**, 53–65 (2018).
8. Arjovsky, M. & Bottou, L. Towards principled methods for training generative adversarial networks. in (arXiv®, 2017).
9. Moen, E. *et al.* Deep learning for cellular image analysis. *Nat. Methods* **16**, 1233–1246 (2019).
10. Weigert, M. *et al.* Content-aware image restoration: pushing the limits of fluorescence microscopy. *Nat. Methods* **15**, 1090–1097 (2018).
11. Belthangady, C. & Royer, L. A. Applications, promises, and pitfalls of deep learning for fluorescence image reconstruction. *Nat. Methods* (2019). doi:10.1038/s41592-019-0458-z
12. Barbastathis, G., Ozcan, A. & Situ, G. On the use of deep learning for computational imaging. *Optica* **6**, 921 (2019).
13. Fang, L. *et al.* Deep Learning-Based Point-Scanning Super-Resolution Imaging. *BioRxiv* (2019). doi:10.1101/740548
14. Flood, N., Watson, F. & Collett, L. Using a U-net convolutional neural network to map woody vegetation extent from high resolution satellite imagery across Queensland, Australia. *International Journal of Applied Earth Observation and Geoinformation* **82**, 101897 (2019).

REVIEWERS' COMMENTS:

Reviewer #2 (Remarks to the Author):

Jin et al. present a machine learning approach to structured illumination microscopy reconstruction. Overall this is an interesting paper and would contribute significantly to the field, allowing reconstructions with lower S:N input data, or fewer input images helping with photobleaching effects which frequently limit super resolution imaging experiments, especially in live cells.

The authors have adequately address almost all my points previously raised and I feel that the paper is now suitable to be published in its current form so long as they address the final issue.

The one outstanding issue I have the claims about the "best" reconstruction method as summarised in supplemental tables 2 and 4. They mark the entries per row with the "best" reconstruction but in almost all cases the differences are completely insignificant given the quoted errors. I think if they wish to mark the "best" algorithm in each case they need to take account of the statistical significance and should mark them as such, say one star for $p < 0.05$, 2 for $p < 0.01$ etc... The exact scheme doesn't matter but they need to clearly mark if the difference between the different models has any likelihood of being real and reproducible.

Minor editing issues:

Bottom of page 3 "...scU-Net provides the least restoration errors...", should be "error"

Page 4 middle,

"...40X less images..." should be "fewer images"

Page 8 Middle:

"...were calculated with NanoJ-NanoJ-SQUIRREL..."

Figure 1 legend: Panel C has box and whisker plots but the authors don't specify what criteria they used to plot the boxes or the whiskers.

Supplementary figure 6 There are 6 different lines, the legend shows how they are group into the high and low light cases, but what are the different lines for, Different experiments, different cells in one experiment?

REVIEWERS' COMMENTS:

Reviewer #2 (Remarks to the Author):

Jin et al. present a machine learning approach to structured illumination microscopy reconstruction. Overall this is an interesting paper and would contribute significantly to the field, allowing reconstructions with lower S:N input data, or fewer input images helping with photobleaching effects which frequently limit super resolution imaging experiments, especially in live cells.

The authors have adequately address almost all my points previously raised and I feel that the paper is now suitable to be published in its current form so long as they address the final issue.

The one outstanding issue I have the claims about the "best" reconstruction method as summarised in supplemental tables 2 and 4. They mark the entries per row with the "best" reconstruction but in almost all cases the differences are completely insignificant given the quoted errors. I think if they wish to mark the "best" algorithm in each case they need to take account of the statistical significance and should mark them as such, say one star for $p < 0.05$, 2 for $p < 0.01$ etc... The exact scheme doesn't matter but they need to clearly mark if the difference between the different models has any likelihood of being real and reproducible.

We understand the reviewer's concerns re reproducibility and have now included all the statistical significance test results in the revised Source Data file, for Supp Table 2 and Supp Table 4. We noticed that in Supp Table 2, U-Net-SIM15 is significantly better than U-Net-SIM3, but the difference is so small that we conclude U-Net-SIM3 could approach the performance of U-Net-SIM15. While in Supp Table 4, in almost all the cases, scU-Net is similar to U-Net-SIM15, but we notice that scU-Net performs better in restoring small missing structures, which might not be reflected by the reporting metrics. ScU-Net also produces better result in live cell reconstruction. We therefore removed the reference to the 'best' algorithm.

Minor editing issues:

Bottom of page 3 "...scU-Net provides the least restoration errors...", should be "error"
Corrected.

Page 4 middle, "...40X less images..." should be "fewer images"
Corrected.

Page 8 Middle: "...were calculated with NanoJ-NanoJ-SQUIRREL..."
Corrected.

Figure 1 legend: Panel C has box and whisker plots but the authors don't specify what criteria they used to plot the boxes or the whiskers.

We added a new **Statistical analysis** subsection under **Methods** to explain the Turkey box-and-whisker plots.

Supplementary figure 6 There are 6 different lines, the legend shows how they are group into the high and low light cases, but what are the different lines for, Different experiments, different cells in one experiment?

Those are different cells in one experiment. We've made it clear in the revised manuscript.